# *Survival of the Most Influential Prompts*:
# Efficient Black-Box Prompt Search via Clustering and Pruning

**Han Zhou[1],*    Xingchen Wan[2],*    Ivan Vulić[1]    Anna Korhonen[1]**

[1]Language Technology Lab, University of Cambridge
[2]Machine Learning Research Group, University of Oxford
{hz416, iv250, alk23}@cam.ac.uk
xwan@robots.ox.ac.uk

## Abstract

Prompt-based learning has been an effective paradigm for large pretrained language models (LLM), enabling few-shot or even zero-shot learning. *Black-box* prompt search has received growing interest recently for its distinctive properties of gradient-free optimization, proven particularly useful and powerful for model-as-a-service usage. However, the discrete nature and the complexity of combinatorial optimization hinder the efficiency of modern black-box approaches. Despite extensive research on search algorithms, the crucial aspect of search space design and optimization has been largely overlooked. In this paper, we first conduct a sensitivity analysis by prompting LLM, revealing that only a small number of tokens exert a disproportionate amount of influence on LLM predictions. Leveraging this insight, we propose the **Cl**ustering **a**nd **P**runing for Efficient Black-box Prompt **S**earch (CLAPS), a simple black-box search method that first clusters and prunes the search space to focus exclusively on influential prompt tokens. By employing even simple search methods within the pruned search space, CLAPS achieves state-of-the-art performance across various tasks and LLMs, surpassing the performance of complex approaches while significantly reducing search costs. Our findings underscore the critical role of search space design and optimization in enhancing both the usefulness and the efficiency of black-box prompt-based learning.

## 1 Introduction

Many of the recent astounding breakthroughs in artificial intelligence have revolved around pretrained large language models (LLMs). Though capabilities of LLMs have advanced at a breakneck speed, modern LLMs are remarkably consistent in that they are almost invariably powered by Transformer-based architectures (Vaswani et al., 2017) pre-

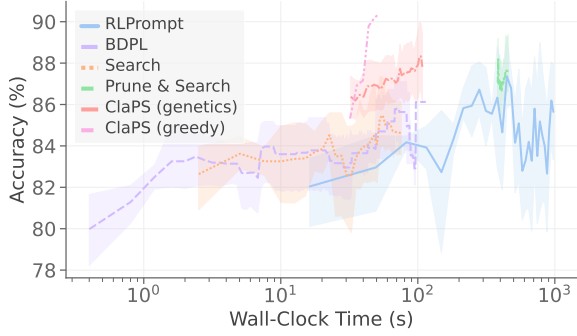

Figure 1: *Our proposed method achieves the best anytime performance*: Anytime test accuracy against wall-clock time CLAPS (our proposed method) compared to other baselines in a few-shot learning setup on SST-2 with Flan-T5$_{base}$. Lines and shades denote the mean and standard deviation over 5 random seeds, respectively (single seed for CLAPS (greedy)).

trained with simple, self-supervised text completion on a large corpus. This is typically followed by fine-tuning and/or, more recently, prompting-based methods on specific tasks (Lyu et al., 2022; Kojima et al., 2022; Chen et al., 2023).

Prompt-based learning is particularly appealing for modern LLMs due to its sample efficiency and flexibility compared to conventional fine-tuning. This enables few-shot or even zero-shot learning (Brown et al., 2020; Liu et al., 2023). It can be categorized into two types: soft and hard prompt tuning. *Soft prompt tuning* directly optimizes the embedding space of the model with other model parameters frozen (Li and Liang, 2021; Lester et al., 2021, inter alia). Although these methods do not require full gradient updates like fine-tuning, they still require parameter access, back-propagation through massive models, and are typically model- and/or task-specific.

*Hard prompt tuning* (HPT), on the other hand, is an emerging paradigm that directly searches for discrete tokens to be added to text input. Hard prompts are more portable and more amenable

---

*Equal contribution. Code is available at https://github.com/cambridgeltl/ClaPS

to human interpretation, as they are actual tokens rather than abstract arrays in the embedding space (Shin et al., 2020). More importantly, unlike soft prompting, which invariably requires parameter access of LLMs due to the need to modify the embeddings, HPT is feasible even if the task LLM is only available as a *'black box'*, i.e., only the model outputs, but not information like parameters and gradients, are available. Indeed, methods leveraging reinforcement learning (RL) (Deng et al., 2022; Zhang et al., 2023) and gradient estimation (Diao et al., 2023) have been recently proposed to exploit this powerful property, particularly since many advanced LLMs (e.g., GPT-4 (OpenAI, 2023) and Bard) are increasingly made available in a model-as-a-service (MaaS) manner, on which parameter or gradient access is expensive or impossible – thus, in this paper, we also focus on this practical but challenging black-box setup.

Despite the promising progress, one challenge plaguing the aforementioned black-box HPT approaches is the difficulty of the discrete and combinatorial optimization inherent to this problem when no gradient guidance is available – it is common that existing methods require a large number of model queries, frequently in the order of $\mathcal{O}(10^3)$ or more, before convergence. While previous works have attempted to alleviate this problem by improving the search strategy, search space design has been largely overlooked. For example, previous works take the natural decision of using the entire tokenizer vocabulary as the search space (Deng et al., 2022), by a convenient extension from the soft prompt tuning. However, as we will show with an analysis of the search spaces of discrete prompts, such a practice is actually suboptimal and has made the optimization unnecessarily difficult. Similar to the phenomenon observed in related discrete optimization problems such as neural architecture search (Wan et al., 2022; Ru et al., 2020; Zhou et al., 2023b), we find the influence exerted by different tokens on the LLM when prepended to the text queries as discrete prompts to be *highly non-uniform*, with a small number of tokens (e.g., 0.1 - 1% of all tokens) exerting a disproportionate amount of influence. Meanwhile, the models are insensitive to or even harmed by the vast majority of the other, 'non-influential' tokens, which nevertheless act as nuisance variables during the search to substantially increase the optimization difficulty and resources required.

Inspired by these findings, we then propose **Cl**ustering **a**nd **P**runing for Efficient Black-box **P**rompt **S**earch (CLAPS), a simple black-box search method that first clusters and prunes the search space to focus on this subset of *influential* tokens, followed by the discrete prompt search on a few-shot objective. We find that after pruning, even the simplest search strategy (e.g., random or evolutionary search) can outperform state-of-the-art methods with much more complicated search strategies, often at a fraction of the search costs over these competing methods (e.g., CLAPS outperforms RLPrompt with only 2.8% of its cost measured in terms of wall-clock time). In summary, in this paper, we offer the following contributions:

**1)** We analyze the influence different tokens in the vocabulary exert on LLM predictions, and find that only a small fraction of tokens positively influence LLMs when used as discrete prompts.

**2)** We propose CLAPS, a black-box discrete prompt search method compatible with a few-shot learning setup, via a cluster-prune-then-search routine that focuses on a small set of influential tokens as discrete prompt candidates.

**3)** We then show that while conceptually simple, CLAPS attains state-of-the-art performance, often achieved at a very small fraction of the cost of competing methods in more than 8 tasks with instruction-finetuned Flan-T5 models.

## 2 Preliminaries

**Hard prompt tuning (HPT).** As mentioned in §1, HPT aims to find *discrete* tokens to be concatenated directly to the test queries with the goal of maximizing task performance. Formally, HPT may be represented as an optimization problem:

$$\mathbf{p}^* = \arg\max_{\mathbf{p} \in \mathcal{P}} \mathbb{E}_{x_i, y_i \sim \mathcal{D}} \Big[ R(\mathbf{f}(C(\mathbf{p}, x_i)), y_i) \Big], \quad (1)$$

where $\{x_i, y_i\}$ denotes a query-target pair, $\mathbf{p} = \{p_1, ..., p_K\}$ are the additional tokens to be concatenated with the text query $x$ – this is often referred to as the *discrete prompts*, whose optimization is the focus of HPT and we use $\mathcal{P}$ to denote the *prompt search space*, the set of all possible discrete prompts. $C(\mathbf{p}, x_i)$ refers to the concatenation of $\mathbf{p}$ and a formatted query $x_i$:

$$C(\mathbf{p}, x_i) = \texttt{Concat}(\mathbf{p}, \texttt{template}(x_i)), \quad (2)$$

where $\texttt{template}(\cdot)$ denotes any human-designed pre-processing procedure that formats $x_i$;

$\mathbf{f}\big(C(\mathbf{p}, x_i)\big) \in \mathbb{R}_{\geq 0}^{|\mathcal{Y}|}$ is the output probability distribution of the model given $x_i$ over all possible classes $\mathcal{Y}$ (defined by the *verbalizers*) with $\sum_{j=1}^{|\mathcal{Y}|} f^{(j)}\big(C(\mathbf{p}, x_i)\big) = 1$; it is worth stressing again that under a *black-box* setup considered in this paper, the output probabilities are the only observation available to us and we assume no access to other information, including but not limited to the model architectures, parameters or gradients. Finally, $R(\cdot, \cdot)$ refers to a *reward function* given the model predictions and the ground-truth labels (an example is the negative cross-entropy loss). The goal of HPT is thus to find the optimal $\mathbf{p}^*$ that maximizes this reward on expectation over some data-generating distribution $\mathcal{D}$. Since the true data-generating distribution is always assumed to be latent, in practice we solve Eq. (1) via empirical risk minimization with a standard train-validation-test split.

**Search strategy and search space.** Solving Eq. (1) is, in general, challenging, as it involves difficult combinatorial discrete optimization, and the gradients essential for standard first-order optimization are not available. A natural recourse, that most previous works have focused on, is developing better zeroth order *search strategies*, via, for example, reinforcement learning and Monte Carlo gradient estimation. *Search space* (i.e., $\mathcal{P}$), on the other hand, is much less well-studied despite the fact that its design has been previously shown to be one of the most important influencing factors in related discrete optimization problems. In HPT, the overall search space $\mathcal{P}$ can be decomposed as a Cartesian product over the search space of individual tokens: $\mathcal{P} = \prod_{k=1}^{K} \mathcal{P}_k$, which is in turn often designed heuristically, and popular choices include the entire tokenizer vocabulary $\mathcal{P}_k = \mathcal{V}$ (and thus $|\mathcal{P}| = |\mathcal{V}|^K$ for a $K$-token discrete prompt) (Deng et al., 2022) or a subset of frequent $n$-grams from it (Diao et al., 2023) – given the exponential scaling w.r.t. the value of $K$, $|\mathcal{P}|$ is typically huge even for modest $|\mathcal{P}_k|$ and/or $K$.

## 3 Analyzing Prompt Search Spaces

**General search spaces are highly redundant.** We argue that, like any other optimization problem, the search space, in our case, may also have a profound effect on both the search strategy and the downstream performance. As the research community of HPT grows, we argue that a systematic

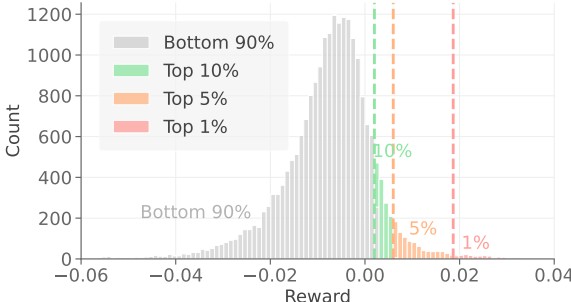

Figure 2: *Only a small fraction of tokens improve performance.* Distribution of the incremental reward $\Delta R(v)$ (Eq. 3) evaluated on 16-shot RTE samples with Flan-T5$_{\text{base}}$. The top-$\{1,5,10\}\%$ tokens in terms of their incremental reward are highlighted in colors.

study of the search space design is crucial. As discussed, existing search spaces are often expensive and heuristically designed. However, a large search space is not necessarily *well-designed*: crucially, it is unknown whether all parts of $\mathcal{P}$ positively contribute to downstream task performance, or it could simply be *highly redundant*, i.e., a large fraction of $\mathcal{P}$ might in fact be unimportant or even harmful, which simply increase complexity but nevertheless act as confounding factors that make the optimization in Eq. (1) unnecessarily hard.

To answer this question, we analyze the building blocks of the most general search space where the individual tokens of the discrete prompts may be any token in the vocabulary $\mathcal{V}$. To quantify the incremental influence for a token $v \in \mathcal{V}$, we define:

$$\Delta R(v) := \frac{\sum_{i=1}^{N} R\big(\mathbf{f}(C(v, x_i)), y_i\big) - R\big(\mathbf{f}(C(x_i)), y_i\big)}{N},$$
(3)

where we treat a token $v$ as a single-token discrete prompt to be concatenated to text queries $x_i$ and its influence $\Delta R(v)$ is the change in reward compared to the case where a formatted input *without* any prompt token $C(x_i)$; $N$ denotes the number of labeled samples randomly sampled from the training set of the target task – we use $N = 16$ throughout this paper, and we define $R(\cdot, \cdot)$ as the negative cross-entropy:

$$R(\mathbf{f}(C(\mathbf{p}, x_i)), y_i) = \sum_{j=1}^{|\mathcal{Y}|} y_i^{(j)} \log f^{(j)}\big(C(\mathbf{p}, x_i)\big). \quad (4)$$

We visualize the results of the above analysis on a representative task in Fig. 2 where we compute $\Delta R(v)$ for all tokens in the vocabulary[1], and we

---

[1]Note that the enumeration here is over $\mathcal{P}_k$, which is typically tractable, as opposed to $\mathcal{P}$. The computation may be further accelerated via clustering – see §4.

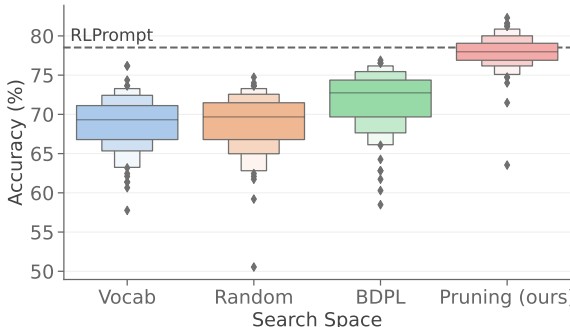

Figure 3: *Pruning improves prompt search.* Distribution of accuracy on RTE with Flan-T5$_{\text{base}}$ by random sampling 100 5-token prompts from different vocabulary spaces. Random refers to a random vocabulary set, and BDPL prunes a context-relevant vocabulary set based on task-dependent $n$-gram scores. Pruning indicates our reward pruning on the vocabulary space. RLPrompt denotes the final test accuracy achieved by RLPrompt (Deng et al., 2022) on this task.

find the distribution of influence over the vocabulary of tokens is, in fact, heavily non-uniform, with a small fraction (roughly 1%, marked in green) of all tokens exerting a disproportionate amount of influence on the prediction of LLMs whereas the vast majority of tokens either actively harm LLM predictions or exert negligible influence.

**Search space pruning.** The finding above means that it would be highly challenging for *any* search method to navigate in the original search space, especially in a black-box setup: the method has to learn to both identify the small fraction of functioning tokens and to avoid the vast majority of unimportant or harmful ones. Instead of doing so, we propose to *prune* $\mathcal{P}_k$ by focusing on the small fraction of the most influential tokens identified above *only* – given the Cartesian structure of $\mathcal{P}$, this results in an exponential reduction of the overall search space $\mathcal{P}$: with a representative $K = 5$ and if we retain the top-1% tokens in terms of $\Delta R(v)$ given by Eq. 3, there is a $\mathcal{O}(10^{10})$ reduction in $|\mathcal{P}|$.

To validate the effectiveness of the pruning procedure and that the search space reduction does not lead to sacrifices in performance, we *randomly* sample 100 5-token discrete prompts from the reduced search space after the aforementioned pruning procedure and use their performances as an approximation of the overall search space quality, and we compare the results against the samples drawn from 1) the original, unmodified search space (Vocab), 2) a reduced search space with $\mathcal{P}_k$ reduced to 10% of the original, but the tokens are

randomly selected (Random), and 3) a $\mathcal{P}_k$ consists of *frequent $n$-grams* selected via pointwise mutual information as in Diao et al. (2023) (BDPL). We visualize the test accuracy distribution in the RTE task in Fig. 3, and we find pruning to massively *improve search space quality* and *reduce search difficulty* compared to both random pruning and the pruning strategy proposed in BDPL, the latter of which does outperform Random and Vocab but is nevertheless outperformed by our pruning strategy. Crucially, the fact that the *median of the 100 randomly sampled discrete prompts already performs similarly to RLPrompt* (Deng et al., 2022), a state-of-the-art method that features much more complicated and expensive RL search strategy and a tailored reward function, highlights the extreme importance of search space design.

## 4 Efficient Black-Box Prompt Search via Clustering and Pruning

Inspired by the analyses presented in §3, we now present Efficient Black-Box Prompt Search via Clustering and Pruning, or CLAPS in short, with the overall procedure illustrated in Fig. 4 and Algorithm 1. At a high level, CLAPS utilizes a multi-step approach, combining the search space pruning proposed in §3 with an optional clustering step to reduce further the computational cost and a simple black-box prompt search routine. We describe the procedure in detail below.

**Clustering.** By default, CLAPS enumerates the tokens in $\mathcal{V}$ and obtains the influence score (Eq. 3)

---

**Algorithm 1** CLAPS.

1: **Input**: Original token search space $\mathcal{P}_k$ (typically the entire vocabulary $\mathcal{V}$); search space size to retain after *clustering* $|\mathcal{V}_c|$ (can be set to $|\mathcal{V}|$ if no clustering is required); search space fraction to retain after *pruning* $\alpha$; discrete prompt length (in terms of # tokens) $K$.
2: **Output**: Optimized discrete prompts $\mathbf{p}^*$
3: **if** $|\mathcal{V}_f| < |\mathcal{V}|$ **then**
4:     **[Clustering]**: Obtain a reduced set of representative tokens $\mathcal{V}_c$ with clustering (§4) as the new token search space $\mathcal{P}_k \leftarrow \mathcal{V}_c$
5: **end if**
6: **for** $v \in \mathcal{P}_k$ **do**
7:     Compute the incremental influence $\Delta R(v)$ of each token $v$ according to Eq. 3.
8: **end for**
9: **[Pruning]**: Rank and prune the tokens in $\mathcal{P}_k$ and only retain the top-$\alpha$ fraction of tokens in terms of $\Delta R(v)$ as the new token search space.
10: **[Search]**: Run black-box prompt search in the *prompt search space* $\mathcal{P} = \prod_k^K \mathcal{P}_k$ to solve Eq. 1 to obtain an optimized discrete prompt $\mathbf{p}^*$.

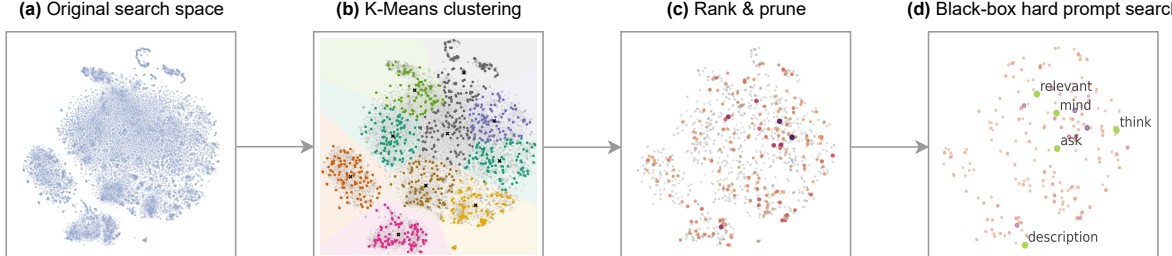

**(a)** Original search space     **(b)** K-Means clustering     **(c)** Rank & prune     **(d)** Black-box hard prompt search

Figure 4: Illustration of the CLAPS pipeline. Starting from **(a)** the original search space (in this case, the entire vocabulary $\mathcal{V}$ with $|\mathcal{V}| \sim \mathcal{O}(10^4)$, visualized via t-SNE plots of vector embeddings *for illustration only*), **(b)** we first perform the optional, unsupervised step of K-Means clustering to retain a fraction of representative tokens $\mathcal{V}_s$ with $|\mathcal{V}_s| \sim \mathcal{O}(10^3)$. We then **(c)** prune the tokens using the procedure described in §3 to retain a small fraction of *influential* ($\sim \mathcal{O}(10^2)$) tokens as the search space. We finally perform **(d)** black-box prompt search over the reduced search space to identify the final $K$-token discrete prompts.

of each token by evaluating on a 16-shot training set. While this procedure, which requires $\mathcal{O}(10^4)$ model evaluations can be already tractable, here we propose an additional optional step to accelerate further our method: instead of enumerating all tokens, we may use an unsupervised algorithm on the token embedding space to obtain a subset of diverse tokens $\mathcal{V}_c$ that well-represent $\mathcal{V}$ (illustrated in Fig. 4(b)) – while alternative methods that explicitly optimize for diversity set selection exist, we opt for the simple greedy K-means++ (Arthur and Vassilvitskii, 2007) to generate $\mathcal{V}_c$ (we set $|\mathcal{V}_c| = 2000$ unless otherwise stated). Formally, for each centroid $\mathbf{e}_c$ identified by K-means++, we collect the closest token in terms of its embedding $\ell_2$ distance:

$$\mathcal{V}_c = \{v_c\}_{c=1}^{|\mathcal{V}_c|} \text{ where } v_c = \arg\min_{v \in \mathcal{V}} \|\mathbf{e}_v - \mathbf{e}_c\|_2. \tag{5}$$

The size of the retained vocabulary $|\mathcal{V}_c|$ is a hyperparameter of the search algorithm (to be discussed in detail at the end of this section) and determines the number of model queries in the next stage, with a smaller $|\mathcal{V}_c|$ leading to more aggressive reduction and improved query efficiency but may lead to some performance loss as some influential tokens may be removed from the search space at this stage. In our experiments, we set $|\mathcal{V}_c| = 2000$ *for all model and task combinations* without further hyperparameter tuning, and after the above procedure, the number of LLM queries at the pruning stage reduces from $\mathcal{O}(10^4)$ to $\mathcal{O}(10^3)$. Empirically, as shown in §6, we find this additional procedure to reduce the cost by roughly 3/4 relative to enumeration (i.e., no pruning) in terms of wall-clock time at only a small performance impact. A sensitivity study of hyperparameters is also performed in §6.

**Ranking and pruning.** As illustrated in Fig. 4(c), we prune $\mathcal{V}_c$ (*with* clustering) or $\mathcal{V}$ (*without* clustering) using the procedure described in §3 to obtain the set of influential tokens for prompt search $\mathcal{V}_{\text{it}}$. The size of $\mathcal{V}_{\text{it}}$ is another hyperparameter, which in this case encodes the *greediness* with a small $|\mathcal{V}_{\text{it}}|$ suggesting a more greedy algorithm that only considers tokens that minimize the validation loss. However, as we empirically show in §6, combining the most influential tokens does not necessarily lead to the optimal prompt, and balancing greediness with prompt search in the next stage leads to the optimal outcome – in this paper, we set $|\mathcal{V}_{\text{it}}| = 200$ for all experiments *without further model- or task-specific hyperparameter tuning*.

**Black-box prompt search.** The final step of CLAPS, as illustrated in Fig. 4(d), is search. To demonstrate that CLAPS is search method-agnostic, we consider three different search strategies in our experiments. To differentiate from previous work focusing on search strategies, we first consider a lightweight search strategy with a basic evolutionary search algorithm with the following ingredients:

- *Initialization*: we initialize with a population of $M$ uniformly sampled $K$-token discrete prompts from the pruned search space, and we evaluate the accuracy of each discrete prompt on a held-out, 16-shot validation set.
- *Evolution*: after evaluating all prompts in the population, at each search epoch, we retain the top 10% of the population in terms of the validation loss as seed prompts. We then generate the next population of $\frac{M}{2}$ prompts via *crossover*, where two randomly selected seed prompts ex-

change tokens to create a new offspring, and $\frac{M}{2}$ new prompts via *mutation*, where we swap a token in a seed prompt with another token in the (pruned) vocabulary with a fixed probability.

- *Termination*: at the end of the final search epoch, we simply return the prompt that leads to the best validation loss seen as the final $\mathbf{p}^*$.

To demonstrate the versatility of CLAPS, we also consider two additional search strategies, namely *greedy search* and *particle swarm optimization* (Kennedy and Eberhart, 1995; Bonyadi and Michalewicz, 2017). The greedy algorithm is a commonly used baseline in combinatorial optimization: Starting with an empty string $\mathbf{p}_0^* := \emptyset$, at the $k + 1$-th iteration, we iterate through the search space $\mathcal{V}_{\mathrm{it}}$ (with $|\mathcal{V}_{\mathrm{it}}| = 200$ following the previous paragraph) and simply select the token that leads to the highest reward, conditioned on partial prompt $\mathbf{p}_{\leq k}^*$ with $k$ tokens already selected so far. More formally, the $(k + 1)$-th token of $\mathbf{p}^*$ is recursively selected by:

$$p_{k+1}^* = \underset{v \in \mathcal{V}_{\mathrm{it}}}{\arg\max} \sum_{i=1}^N R(\mathbf{f}(C(\mathrm{Concat}(\mathbf{p}_{\leq k}^*, v), x_i)), y_i),$$
(6)

and the algorithm terminates when all $K$ tokens are selected. For the particle swarm optimizer, we use an adapted version of the algorithm described by Zang et al. (2020) to work in the discrete search space, and we refer the reader to Appendix A for further implementation details.

It is worth noting that we only consider a small representative, and definitely non-exhaustive set of search algorithms. CLAPS, which focuses on search space design, can be deemed as a *meta*-method that is compatible with any search strategy, including but not limited to the ones proposed in previous work, in a *plug-and-play* manner. It is therefore possible that combining CLAPS with a more advanced search method would lead to even stronger performance – we defer a thorough investigation to future work.

## 5 Related Work

**Prompt learning.** Prompt learning is a class of powerful methods for LLM adaptation and has become an efficient alternative to full model finetuning (Liu et al., 2023). Earlier methods (Li and Liang, 2021; Lester et al., 2021; Liu et al., 2022b) typically feature *soft* prompt tuning, where continuous prompts which modify the input embedding of an otherwise frozen LLM are optimized.

Other methods, such as the *parameter-efficient fine-tuning* (PEFT) techniques (He et al., 2022), which only tune a small fraction of the model parameters (Houlsby et al., 2019; Hu et al., 2022), may also be regarded as soft prompt learning. While promising, a drawback of the soft prompting methods is that since the model-specific input embedding layers often need to be modified, these methods inevitably require internal model access. Furthermore, with a few exceptions like BBT (discussed in the next paragraph), many soft prompting methods still require back-propagation of gradients through massive models, which can still be computationally expensive. In contrast to soft prompting, *hard* prompt learning learns discrete tokens: AutoPrompt (Shin et al., 2020) uses model gradients to select appropriate tokens automatically, but is nevertheless restricted to a 'white-box' setup.

**Black-box prompt optimization.** In contrast to the white-box methods discussed above, several methods are proposed to tune discrete prompts in a black-box manner (i.e., not using internal knowledge about the pretrained LLM). Black-box tuning (BBT) and BBTv2 (Sun et al., 2022b,a) use gradient-free optimization to learn soft prompts that are projected back to the embedding/weight space and concatenated to the query embedding and/or weights. While not using model gradients, these methods nevertheless require access to input embedding *of the task model itself*, and hence are not black-box in the strictest sense. In the strictly black-box setup, methods using reinforcement learning (Deng et al., 2022; Zhang et al., 2023), discrete optimization (Prasad et al., 2023), and gradient estimation (Diao et al., 2023) have been proposed; we empirically compare against them in §6. Furthermore, as discussed in §4, CLAPS is fully orthogonal to the previous work since these techniques focus on improving *search strategy*. Several other works have focused on optimizing specific components of the prompt design, e.g., Rubin et al. (2022); Liu et al. (2022a); Wan et al. (2023a,b) focus on selecting in-context examples and Zhou et al. (2023a) mitigate the in-context bias by calibration. We argue that these methods are again orthogonal to our contributions and thus may offer combining benefits.

## 6 Experiments and Results

**Evaluation data.** We include various tasks from single-sentence to multi-sentence classification

| Model | Flan-T5$_{base}$ | | | | | CLAPS | | Flan-T5$_{large}$ | | | | | CLAPS | |
|---|---|---|---|---|---|---|---|---|---|---|---|---|---|---|
| Method | FT | Manual | BDPL | RLP. | Search | Genetics | Greedy | FT | Manual | BDPL | RLP. | Search | Genetics | Greedy |
| SST-2 | 76.19 | 85.32 | 84.89 | 86.01 | 85.85 | **87.78** | **90.37** | 83.72 | 92.32 | 92.43 | 92.55 | 92.73 | **93.03** | **94.27** |
| RTE | 51.55 | 73.65 | 72.27 | 78.52 | 77.47 | **81.23** | **79.42** | 49.10 | 84.12 | 84.12 | 84.55 | 85.05 | **86.12** | **86.28** |
| SNLI | 60.98 | 48.97 | 51.65 | 63.06 | 59.30 | **65.92** | **63.47** | 74.62 | 76.50 | 78.05 | **85.57** | 84.27 | 84.08 | **84.75** |
| QNLI | 67.94 | 62.40 | 61.53 | **74.85** | 65.83 | 70.52 | **80.07** | 77.32 | 82.67 | 80.45 | 83.80 | 82.78 | **85.81** | **86.47** |
| MNLI | 45.99 | 43.15 | 42.52 | **57.60** | 47.30 | 50.45 | **57.02** | 51.91 | 70.68 | 75.32 | **80.85** | 79.26 | **81.81** | 77.82 |
| MRPC | 68.73 | 69.12 | **71.49** | 58.82 | **72.74** | 68.43 | 65.93 | 70.83 | 76.72 | **83.96** | 80.15 | 74.85 | **77.11** | 75.49 |
| QQP | 66.31 | 79.07 | 68.44 | 80.09 | 80.57 | **80.87** | **81.40** | 77.23 | 81.29 | 77.25 | 72.23 | **82.01** | **81.31** | 78.10 |
| News | **83.62** | 71.15 | 70.71 | 76.91 | **77.20** | 76.06 | 77.13 | **83.72** | 81.22 | 80.77 | 82.62 | 82.96 | **84.24** | 83.08 |
| Average | 65.16 | 66.60 | 65.44 | 71.98 | 70.78 | **72.66** | **74.35** | 71.06 | 80.69 | 81.54 | 82.39 | 82.99 | **84.19** | 83.28 |

Table 1: Accuracy on Flan-T5$_{base}$ (**Left**) and Flan-T5$_{large}$ (**Right**). We reproduce all baselines and report the mean for 5 random seeds for Flan-T5$_{base}$. For computation-expensive experiments, we report single-seed results for Flan-T5$_{large}$. The **best** and second-best results are marked in bold fonts and ranked by color.

| Model | Flan-T5$_{base}$ | | | | | CLAPS | | Flan-T5$_{large}$ | | | | | CLAPS | |
|---|---|---|---|---|---|---|---|---|---|---|---|---|---|---|
| Method | FT | Manual | BDPL | RLP. | Search | Genetics | Greedy | FT | Manual | BDPL | RLP. | Search | Genetics | Greedy |
| BG | 34.85 | 33.97 | 33.36 | **37.16** | 35.67 | **36.16** | 34.57 | 33.95 | 43.87 | **44.91** | 44.61 | 44.78 | **45.13** | 44.63 |
| DE | 37.35 | 35.47 | 35.07 | 38.16 | 36.91 | **39.56** | **41.64** | 40.56 | 55.17 | 55.63 | **62.67** | 60.80 | **62.29** | 57.74 |
| EN | 47.69 | 41.20 | 40.08 | **54.23** | 46.17 | 54.14 | **54.77** | 69.04 | 69.84 | 76.77 | **81.18** | 79.04 | **81.15** | 77.11 |
| ES | 35.51 | 34.53 | 33.82 | 34.27 | 36.62 | **38.00** | **40.50** | 52.40 | 53.03 | 51.92 | **58.58** | 58.55 | **60.52** | 55.59 |
| FR | **40.23** | 35.15 | 33.92 | 37.27 | 35.67 | 38.16 | **41.98** | 48.30 | 54.69 | 55.07 | **63.81** | 60.35 | **63.76** | 57.62 |
| HI | 33.50 | 33.33 | 33.33 | 33.13 | 33.51 | **33.99** | **34.61** | 32.95 | 33.81 | 34.41 | **35.45** | 34.97 | **35.03** | 34.61 |
| RU | 34.99 | 33.85 | 33.42 | 33.51 | 36.91 | **37.46** | **37.01** | 37.13 | 48.96 | 49.34 | **49.74** | 49.68 | **50.62** | 47.84 |
| SW | 34.01 | 33.37 | 33.33 | **34.19** | 33.72 | 33.71 | **35.95** | 36.33 | 36.87 | 37.29 | **38.94** | 36.67 | **37.56** | 36.81 |
| TR | 34.14 | 33.81 | 33.54 | **36.53** | 33.86 | 36.11 | **36.67** | 35.35 | 43.91 | 44.57 | **45.69** | 45.85 | 45.31 | 42.91 |
| Avg. | 36.92 | 34.96 | 34.43 | 37.61 | 36.56 | **38.59** | **39.74** | 42.89 | 48.91 | 50.00 | **53.41** | 52.30 | **53.49** | 50.54 |

Table 2: Accuracy in 9 languages on XNLI with Flan-T5$_{base}$ (**Left**) and Flan-T5$_{large}$ (**Right**). Methods in other languages show similar performance to Hindi and Swahili with marginal improvements over random prediction and are omitted from the table. We report single-seed RLPrompt results due to its computation costs for XNLI tasks. Refer to Table 1 for additional explanations.

tasks, from mono-lingual to multi-lingual NLI datasets for widely validating the performance of CLAPS at different levels of task difficulty. We conduct experiments on the standard GLUE dataset (Wang et al., 2018) including: SST-2, RTE, QNLI, MNLI, MRPC, QQP. Furthermore, we include AG's News (Zhang et al., 2015) and SNLI (Bowman et al., 2015) following the previous hard prompt tuning papers (Deng et al., 2022; Zhang et al., 2023). In addition, we include XNLI (Conneau et al., 2018), a multilingual NLI task, as the most challenging unseen dataset for revealing the potential of our method in different languages. For all tasks, we follow the standard few-shot setting (Perez et al., 2021), where 16 shots represent 16 examples per class for both training and validation sets. Since the test labels for GLUE tasks are unavailable, following standard practice we take validation shots from the training sets and treat the validation set as the test set.

**Baselines.** In the few-shot learning setup, we mainly compare CLAPS with gradient-free black-box baselines. Training details of all the methods

in comparison are included in Appendix A.

• *Finetuning:* we finetune the seq2seq language model following the standard setup (Karimi Mahabadi et al., 2021; Zeng et al., 2023).

• *BDPL* (Diao et al., 2023): BDPL first models the prompt generation as samples drawn from a multi-dimensional categorical distribution, and uses Monte Carlo-estimated gradients to optimize the distribution parameters. The search space is over a subset of $\mathcal{V}$ that appear as frequent $n$-grams in the task training corpus.

• *RLPrompt* (Deng et al., 2022): It trains a policy network that generates discrete prompts (an MLP layer on top of a frozen, pretrained GPT-2 model) with a bespoke piece-wise reward. The search space is over the whole vocabulary.

• *Search* and *Prune & Search*: we include these baselines both as ablation experiments and to directly gauge the impact of search space design on the downstream task performance. *Search* baseline utilizes the genetics search algorithm described in §4 directly in the full, non-pruned vocabulary

search space without clustering and pruning. *Prune & Search* refers to CLAPS without the clustering step where we prune on the whole vocabulary followed by the genetics search.

**Models.** We explore the potential of CLAPS with instruction-finetuned models, and we test on a wide range of challenging tasks with Flan-T5$_{base}$ and Flan-T5$_{large}$ models, one of the most powerful open-sourced models of their size (Chung et al., 2022). We refer readers for detailed hyperparameters and training setups to Appendix A.

**Discussion of main results.** We present the results on all tasks except XNLI in Table 1, whereas the XNLI results are provided in Table 2. For CLAPS results, we present CLAPS with genetics and greedy search algorithms in the main text and show the results with particle swarm optimization in Appendix B. Across both sets of tasks, we find CLAPS (i) to consistently improve on standard, no-prompt *Manual* baseline and (ii) to outperform the other prompting baselines across the models and tasks. More specifically, CLAPS (genetics) outperforms RLPrompt 0.6% and 1.8% on average for Flan-T5$_{base}$ and Flan-T5$_{large}$, respectively. In addition, we find that when used with CLAPS, the greedy search algorithm, although straightforward, can be surprisingly strong across many experiments except for XNLI with Flan-T5$_{large}$; this concretely shows that CLAPS may orthogonally benefit different suitable search algorithms. Furthermore, in contrast to other prompting baselines like BDPL and RLPrompt, which occasionally lead to performance deterioration from *Manual*, CLAPS consistently improves over the latter. We hypothesize that it is exactly due to the stability of our approach enabled by searching only on pruned search space featuring positively influential tokens, whereas the competing methods may suffer from unstable and noisy gradient estimations and/or RL policies over a search space with more harmful sub-components.

Finally, we emphasize that CLAPS achieves state-of-the-art performance with rather naïve search strategies, which stands in stark contrast to the competing methods that are both much more complicated methodologically and often orders-of-magnitude more expensive – we argue that this highlights that methods focusing on search space design warrant further investigation in future work.

**Efficiency analysis.** We analyze the performance-cost trade-off of various methods on a represen-

| Methods | # Param. | VRAM | Time | # Query | SST-2 (%) |
|---|---|---|---|---|---|
| FT | 250M | 6.53GB | 0.42 min | - | 76.15 |
| RLPrompt | 3M | 3.60GB | 65.1 min | 12000 | 86.01 |
| BDPL | 1K | 2.54GB | 0.20 min | 600 | 84.89 |
| Search | 0 | 2.54GB | 1.26 min | 4000 | 85.82 |
| Prune & Search | 0 | 2.54GB | 7.65 min | 24000 | 87.84 |
| CLAPS (Genetics) | 0 | 2.54GB | 1.80 min | 6000 | 87.78 |
| CLAPS (Greedy) | 0 | 2.54GB | 0.86 min | 3000 | 90.37 |

Table 3: Comparing the efficiency of CLAPS with baselines in the few-shot learning setup with Flan-T5$_{base}$. We report the number of trainable parameters, the peak VRAM load, and the wall-clock time for the training phase of all methods. The pruning-phase time span is included in CLAPS. Note that RLPrompt and BDPL are run under their default settings, respectively.

| Dataset | RLPrompt-found Prompt |
|---|---|
| SST-2 | ReviewCustomerBankBankBank |
| RTE | DatabaseansweranswerYesĜyes |
| QQP | ComponentArgsArgsChangeĜaffecting |
| XNLI$_{EN}$ | NodeArgsArgsArgsĜaffecting |

| Dataset | CLAPS-found Prompt |
|---|---|
| SST-2 | cruise perfect properly review cruise |
| RTE | answer respectively minimum tell answer |
| QQP | suggest outside cause exists statement |
| XNLI$_{EN}$ | think ask relevant description mind |

Table 4: Examples of CLAPS-discovered prompts compared to RLPrompt for a collection of tasks using Flan-T5$_{base}$. The CLAPS prompts are prepended to the formatted text queries, whose templates are listed in Appendix A.

tative task in Table 3, which highlights the much-enhanced practicality of CLAPS compared to the baselines: CLAPS is extremely *storage*-efficient as it requires no additional parameters to be stored in the GPU, and the only memory requirement is to maintain the task model under an inference-only (i.e., no gradient storage) mode. CLAPS also achieves the best trade-off between *time* efficiency and performance as faster methods (FT and BDPL) perform much worse, whereas methods like RL-Prompt perform better, but are orders-of-magnitude slower. It is worth noting for fairness of comparison, we also perform additional experiments by running BDPL longer than the default, but we find that doing so only brings marginal improvement over the *Manual* baseline, as illustrated in Fig. 1.

**Examples of discovered discrete prompts.** Table 4 presents examples of CLAPS-discovered prompts, and interestingly, we often observe some interpretability even though CLAPS has not been explicitly tuned towards fluency. For example, in SST-2, a movie review sentiment-classification task, CLAPS picks 'review' as a part of the best

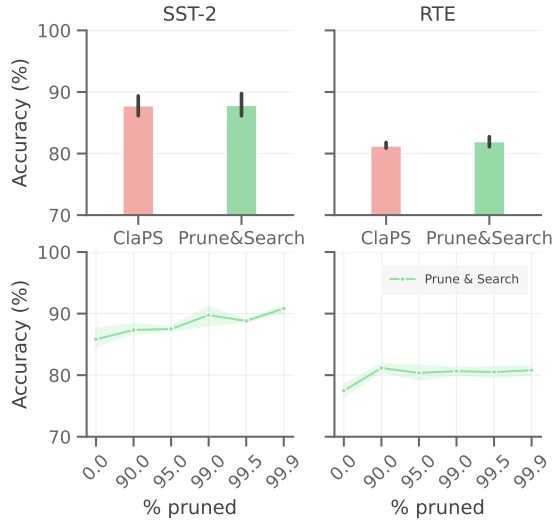

Figure 5: Ablation studies. **Top row**: Flan-T5$_{base}$ accuracy on SST-2/RTE *with* (`ClaPS-genetics`) and *without* (`Prune&Search`) clustering. **Bottom row**: performance sensitivity to pruning strength from 0% (no pruning, i.e., the *Search* baseline in Tables 1 & 2 to 99.9%. Mean ± standard deviation (error bar or shade) shown.

prompt. On the other hand, RTE and XNLI$_{EN}$ are both textual entailment tasks and CLAPS again spontaneously discovers prompts provide an instruction-like signal to 'ask' the model to 'answer' the question. While the other prompts are less immediately interpretable, we hypothesize that they nevertheless act to tune the model embedding towards the optimal direction for the target task for performance improvement. CLAPS does share some words with the competitive baseline, RLPrompt, and these words (e.g., 'review' and 'answer') are usually 'influential prompts' identified by our pruning strategy and have significant impacts on the model's prediction. With a similar or even better quality of prompts, CLAPS stands out by first establishing an efficient search space while saving substantial computation costs.

**Ablation and sensitivity studies.** In Fig. 5, we first study the performance impact of the use of clustering by comparing CLAPS against *Prune&Search*: we find that in the tasks considered, clustering minimally affects the performance, but leads to a ~75% speed-up in terms of wall-clock time. We also investigate the effect of different pruning strengths, and find that 1) pruning generally improves performance, 2) performance is rather insensitive to (reasonable) pruning strength, and 3) the threshold of 1% (corresponding to 99% in Fig. 5) is a generalizable choice across tasks. Finally, we conduct

additional ablation experiments to test the robustness of CLAPS w.r.t. other hyperparameters, such as the number of clusters during clustering and the prompt length; the readers are referred to Appendix B for details.

# 7 Conclusion

We first analyzed the search spaces in the general paradigm of hard prompt search. Inspired by the findings that only a small fraction of tokens exert a positive influence on prediction, we proposed CLAPS, an efficient black-box prompt search method via clustering and pruning. The CLAPS method is methodologically simple, easy to implement, and cost-effective, and we showed that it achieves state-of-the-art performance in both mono-lingual and multi-lingual tasks with Flan-T5 models. CLAPS is a meta-method orthogonal to the search strategy, and we expect more efficient and effective prompt search algorithms can be created on top of it. We hope that future work will invest more time into the important problem of search space design.

## Limitations

We argue that CLAPS only serves as a first step towards the promising direction of better search space design and automation, and thus, the room for improvement is ample. First, we have only considered a suite of natural language understanding (NLU) tasks that may be cast as classification in the present study, whereas prompting techniques for generative tasks are, in general, less developed.

Second, we have only explored a token-based search space for hard prompts as it is the most general, but alternative search spaces built on the overall instruction templates and exemplifiers exist (such as the ones used in Zhang et al. (2023) and Prasad et al. (2023). We hypothesize that since these search spaces are also often heuristically designed, the search space issues and the pruning procedure may also apply to these search spaces, which are often claimed to be more interpretable, and thus, it would be interesting to extend our analysis, and methodology to these alternative spaces.

Third, as we discussed in §4, while the present paper primarily focuses on search space, it is possible to combine CLAPS with more advanced search methods for further potential gains: some promising strategies include reinforcement learning, as used in Deng et al. (2022) and Zhang et al. (2023),

and sample-efficient zeroth-order algorithms that may operate directly over the token search spaces, such as the recent advancements in Bayesian optimization over discrete and/or combinatorial variables (Baptista and Poloczek, 2018; Wan et al., 2021; Daulton et al., 2022). We defer thorough investigations to future work.

## Acknowledgements

Han Zhou is supported by the UK Research and Innovation (UKRI) Frontier Research Grant EP/Y031350/1 (the UK government's funding guarantee for ERC Advanced Grants) awarded to Anna Korhonen at the University of Cambridge. Xingchen Wan is supported by the Clarendon Scholarship at University of Oxford. The work has been supported in part by a Royal Society University Research Fellowship (no 221137; 2022-) awarded to Ivan Vulić, and by the UK EPSRC grant EP/T02450X/1.

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

## A Implementation Details

### A.1 Additional Experimental Details

For all experiments that record wall-clock time, we run on a single RTX 4090 24GB GPU. The main experimental results in Table 1 & 2, we run on the RTX 4090 24GB GPU and the A100 80GB GPU.

**CLAPS.** During the phase of search space pruning, we exclusively focus on tokens with a space in front of it, which removes the majority of tokens that are not a single word and various symbols across different languages. In clustering, we collect 2000 centroids with the closest word in the embedding space. Then, we filter the clustered set by removing repetitive words, which then gives 1867 tokens as the initial search space before pruning.

In implementing the evolutionary search algorithm, we conduct a 30-epoch search with a population size of 128, and both mutation and crossover size of 64. At each epoch, we retain a 10% fraction of top candidates to the next epoch. We searched only for the 5-token length for all our experiments.

For the particle swarm optimization, we use the open-source implementation (https://github.com/thunlp/SememePSO-Attack) from Zang et al. (2020) that is compatible with discrete search spaces over word tokens. The key changes that we made were to reflect the fact that the prompting setup is less restrictive than the adversarial attack that Zang et al. (2020) considered, which would also require the changes in text to be as imperceptible as possible. As such, we removed constraints such as only substitution is allowed, and, unlike adversarial attacks where the algorithm is terminated whenever a successful perturbation is found, we always run full 40 epochs.

**Fine-tuning.** We follow the same implementation of standard T5 fine-tuning (Karimi Mahabadi et al., 2021; Zeng et al., 2023). Due to the simplicity of the seq2seq pipeline, we keep the default training templates for all fine-tuning experiments.

**BDPL.** We use the same prompt templates for both CLAPS and BDPL. We use the default hyperparameter setup for all its experiments with a sample size of 20 and 30 epochs. For the experiments in Fig. 1, we run BDPL for roughly the same wall-clock time as CLAPS, which lasts 360 epochs.

**RLPrompt.** We implement the same prompt templates for both CLAPS and RLPrompt for a fair comparison. We report the default hyperparam-

| Method | FT | Manual | BDPL | RLP. | Search | CLAPS | | |
|---|---|---|---|---|---|---|---|---|
| | | | | | | Genetics | Particle Swarm | Greedy |
| SST-2 | $76.19_{0.93}$ | 85.32 | $84.89_{2.29}$ | $86.01_{1.32}$ | $85.85_{1.79}$ | $\mathbf{87.78_{1.77}}$ | $87.55_{1.81}$ | $\mathbf{90.37}$ |
| RTE | $51.55_{2.46}$ | 73.65 | $72.27_{2.37}$ | $78.52_{0.60}$ | $77.47_{1.13}$ | $\mathbf{81.23_{0.56}}$ | $79.93_{1.85}$ | 79.42 |
| SNLI | $60.98_{4.18}$ | 48.97 | $51.65_{3.95}$ | $63.06_{0.42}$ | $59.30_{2.27}$ | $65.92_{3.14}$ | $\mathbf{66.12_{3.35}}$ | 63.47 |
| QNLI | $67.94_{4.19}$ | 62.40 | $61.53_{1.81}$ | $\mathbf{74.85_{3.40}}$ | $65.83_{2.20}$ | $70.52_{2.04}$ | $69.45_{1.73}$ | $\mathbf{80.07}$ |
| MNLI | $45.99_{4.81}$ | 43.15 | $42.52_{4.29}$ | $\mathbf{57.60_{0.88}}$ | $47.30_{3.85}$ | $50.45_{3.09}$ | $53.71_{1.78}$ | $\mathbf{57.02}$ |
| MRPC | $68.73_{0.71}$ | 69.12 | $\mathbf{71.49_{7.93}}$ | $58.82_{1.83}$ | $\mathbf{72.74_{1.11}}$ | $68.43_{3.03}$ | $70.83_{2.14}$ | 65.93 |
| QQP | $66.31_{2.81}$ | 79.07 | $68.44_{2.64}$ | $80.09_{0.59}$ | $80.57_{1.76}$ | $80.87_{0.75}$ | $\mathbf{81.51_{0.25}}$ | $\mathbf{81.40}$ |
| AG's News | $\mathbf{83.62_{0.77}}$ | 71.15 | $70.71_{0.60}$ | $76.91_{0.76}$ | $\mathbf{77.20_{1.76}}$ | $76.06_{0.86}$ | $77.03_{1.13}$ | 77.13 |
| Average | 65.16 | 66.60 | 65.44 | 71.98 | 70.78 | 72.66 | $\mathbf{73.27}$ | $\mathbf{74.35}$ |

Table 5: Accuracy on Flan-T5$_{base}$ with three different search algorithms on CLAPS. We reproduce all baselines and report the mean and standard deviation for 5 random seeds for Flan-T5$_{base}$. The **best** and second-best results are marked in bold fonts and ranked by color.

| #Cluster | 20000 | 6000 | 2000 | 1000 |
|---|---|---|---|---|
| SST-2 | $87.84_{2.11}$ | $87.66_{2.07}$ | $87.78_{1.77}$ | $88.19_{0.52}$ |
| RTE | $81.95_{0.94}$ | $80.43_{1.57}$ | $81.23_{0.56}$ | $77.91_{1.43}$ |

Table 6: Performance of CLAPS (Genetics) with respect to the number of clusters in the phase of clustering.

| #Token | 2 | 5 | 10 |
|---|---|---|---|
| SST-2 | $87.04_{1.18}$ | $87.78_{1.77}$ | $87.41_{0.88}$ |
| RTE | $80.07_{1.87}$ | $81.23_{0.56}$ | $79.49_{1.59}$ |

Table 7: Performance of CLAPS (Genetics) over the number of discrete tokens in the stage of black-box prompt search.

eter setup for all experiments. For computationally expensive experiments with XNLI or using Flan-T5$_{large}$ as the backbone, we set the training steps as 6000 instead of 12000. In addition, since RLPrompt requires an order-of-magnitude training cost than CLAPS, we set a strict wall-clock limit to all RLPrompt experiments that go beyond 12 training hours via A100. We then take the same number of evaluation prompts as CLAPS at fixed time intervals. Following RLPrompt's open-source script, we test the prompt with the highest validation reward score.

## B  Additional Experimental Results

We attach the main experimental results from Table 1 with standard deviation and one additional CLAPS results by particle swarm optimization in Table 5. Based on three different CLAPS search strategies, we find that, in absolute terms, it does matter what search strategy is used to yield improved task performance, and this is thus largely task-dependent. In relative terms, CLAPS improves almost all of the tasks with significantly enhanced efficiency, validating its orthogonality to the selected search algorithm.

We provide an ablation study that conducts the CLAPS pipeline with different numbers of clusters in the phase of clustering in Table 6. It reveals that having the number of clusters at 2000 stands as a good empirical trade-off point for saving the cost while showing strong performance across tasks.

We then provide an ablation study with various token lengths in Table 7. First, increasing the token length from 2 to 5 leads to an improvement in performance over the two tasks. It shows that a longer sentence can provide more expressive control and description for prompting the language model. By further increasing the token length from 5 to 10, we observe a decrease in performance and consider this to be due to the dimensionality problem in derivative-free optimization.

## C  Prompt Template

We present the prompt template of the tasks considered in Table 8.

| Dataset | Prompt template |
|---------|-----------------|
| SST2 | Template: {prompt}. Sentence: {sentence$_1$}, Sentiment: |
|  | Verbalizer: negative, positive |
|  | CLAPS: `cruise perfect properly review cruise` |
| RTE | Template: {prompt}. Sentence 1: {sentence$_1$}, Sentence 2: {sentence$_2$}, Textual Entailment: |
|  | Verbalizer: yes, no |
|  | CLAPS: `answer respectively minimum tell answer` |
| SNLI | Template 1: {prompt} {sentence$_1$} {sentence$_2$} Entailment: |
|  | Template 2: {prompt}. In this task, the goal is to predict textual entailment with 'yes' 'maybe' 'no'. sentence A implies sentence B entailment: yes; sentence A is neutral to sentence B entailment: maybe; sentence A contradicts sentence B entailment: no. Sentence A: {sentence$_1$}, Sentence B: {sentence$_2$}, Entailment: |
|  | Verbalizer: yes, maybe, no |
|  | CLAPS: `möchten kannst procent dass that` |
| QNLI | Template: {prompt}. Question: {sentence$_1$}, Sentence: {sentence$_2$}, Entailment: |
|  | Verbalizer: yes, no |
|  | CLAPS: `leider respectively read grey respectively` |
| MNLI | Template 1: {prompt} {sentence$_1$} {sentence$_2$} Entailment: |
|  | Template 2: {prompt}. In this task, the goal is to predict textual entailment with 'yes' 'maybe' 'no'. sentence A implies sentence B entailment: yes; sentence A is neutral to sentence B entailment: maybe; sentence A contradicts sentence B entailment: no. Sentence A: {sentence$_1$}, Sentence B: {sentence$_2$}, Entailment: |
|  | Verbalizer: yes, maybe, no |
|  | CLAPS: `tell relevant statement suggest suggest` |
| MRPC | Template: {prompt}. Sentence 1: {sentence$_1$}, Sentence 2: {sentence$_2$}, Semantically Equivalent: |
|  | Verbalizer: no, yes |
|  | CLAPS: `courses beschrieben serial vertical Über` |
| QQP | Template: {prompt}. Sentence 1: {sentence$_1$}, Sentence 2: {sentence$_2$}, Semantically Equivalent: |
|  | Verbalizer: no, yes |
|  | CLAPS: `suggest outside cause exists statement` |
| AG's News | Template: {prompt}. Classify the news articles into the categories of World, Sports, Business, and Technology. {sentence$_1$}: |
|  | Verbalizer: World, Sports, Business, Technology |
|  | CLAPS: `prize computing panel Congress certified` |
| XNLI | Template 1: {prompt} {sentence$_1$} {sentence$_2$} Entailment: |
|  | Template 2: {prompt}. In this task, the goal is to predict textual entailment with 'yes' 'maybe' 'no'. sentence A implies sentence B entailment: yes; sentence A is neutral to sentence B entailment: maybe; sentence A contradicts sentence B entailment: no. Sentence A: {sentence$_1$}, Sentence B: {sentence$_2$}, Entailment: |
|  | Verbalizer: yes, maybe, no |
|  | CLAPS: `think ask relevant description mind` |

Table 8: Prompt templates for prompt search experiments, where we also implement the same template for both BDPL and RLPrompt experiments. We use `Template 1` for Flan-T5$_{\text{base}}$ experiments. Due to the level of difficulty of SNLI/MNLI/XNLI, we evaluate `Template 2` with task instruction for Flan-T5$_{\text{large}}$ experiments, which demonstrate better in-context learning capability than small language models. All templates are manually created and fixed without iterations.