# OpenReview forum: "Survival of the Most Influential Prompts: Efficient Black-Box Prompt Search via Clustering and Pruning"
_EMNLP/2023/Conference — EMNLP 2023 Findings_

### Official Review · Reviewer_3YDq · 2023-08-05

**Soundness:** 4

**Excitement:**

2: Mediocre: This paper makes marginal contributions (vs non-contemporaneous work), so I would rather not see it in the conference.

**Paper Topic And Main Contributions:**

- This paper deal with the problem of black-box discrete prompt optimization.
- Existing black-box discrete prompt optimization papers consider catesian product of whole token sets as the search space.
- On the other hand, the authors propose ClaPS that search prompt on the reduced search space. In detail, they cluster tokens on the embedding space and prune redundant tokens to construct reduced candidates of tokens.
- The empirical results show that ClaPS achieve state-of-the-art peformance with faster runtime compared to baseline methods.

**Questions For The Authors:**

- Discrete prompt optimization problem is a combinatorial optimization problem. In ClaPS, the authors propose search algorithm based on evolutionary method. However, the effectiveness of the proposed search method was not exploited enough. I list some possible candidates of method.
1. Greedy : Similar to ClaPS, [1] reduce the token cadidate set using gradient signal and search optimal prompt using greedy algorithm. I think search algorithm can be directly applied into the setting of ClaPS.
2. Bayesian Optimization : Adversarial attack research on NLP models also solves similar combinatorial optimization problem with discrete prompt optimization. [2] uses Bayesian optimization (BO) as the solver.
3. Particle swarm optimization : [3] uses particle swarm optimization (PSO) as the solver.
4. RLPrompt : I think you may directly plug in RLPrompt method as the search method for the reduced search space.
If possible, can you show the superiority of your proposed search algorithm compared to some possible candidates of search algorithm? Also, I want to see the curve of validation accuracy and test accuracy for each optimization method.
- Since ClaPS is a black-box prompt optimization method, you may apply the proposed method into black-box API such as GPT-4 or ChatGPT. Can you provide comparison between manual prompt and discrete prompt optimized by ClaPS?
- In the black-box setting, the number of queries to the target model is one of the most important factor that measures computational resource. Can you provide test accuracy vs number of queries plot of ClaPS and other black-box prompt optimization method? It will help visualize query-efficiency of your method.
- If I understood correctly, ClaPS require some queries on target model during clustering and pruning procedure (for ranking candidates). Does the resource (time and vram) in table 3 contain the resource used in clustering and pruning step?

I promise that I'll raise my score if the responses are satisfactory.

[1] AutoPrompt: Eliciting Knowledge from Language Models with Automatically Generated Prompts, Shin et al., EMNLP 2020

[2] Query-Efficient and Scalable Black-Box Adversarial Attacks on Discrete Sequential Data via Bayesian Optimization, Lee et al., ICML 2022

[3] Word-level Textual Adversarial Attacking as Combinatorial Optimization, Zang et al., ACL 2020

[4] RLPrompt: Optimizing Discrete Text Prompts with Reinforcement Learning, Deng et al., EMNLP 2022

**Reasons To Accept:**

- The existing discrete black-box prompt optimization methods suffer from computational cost caused by their huge search space. The proposed method solve this issue based on clustering and pruning the token candidates to search.
- Sequentially, ClaPS solves the problem of high computational cost in existing methods such as RLPrompt.

**Reasons To Reject:**

- However, I'm curious whether the proposed search algorithm is suitable algorithm to search in the reduced search space. I think ablation on search algorithm is necessary to make sure that the algorithm is one of the contribution or not.
- Also, the experiments are only conducted on Flan-T5 models.
- The number of queries to the target model should be considered as the evaluation metric for resource usage.

Please refer to questions.

**Reproducibility:**

3: Could reproduce the results with some difficulty. The settings of parameters are underspecified or subjectively determined; the training/evaluation data are not widely available.

**Reviewer Confidence:**

4: Quite sure. I tried to check the important points carefully. It's unlikely, though conceivable, that I missed something that should affect my ratings.

---

> ### Author Rebuttal · Authors · 2023-08-29
>
> We thank the reviewer for their insightful comments, especially the suggestions on additional search methods, which are particularly useful in further strengthening our manuscript. We refer the reviewer for a detailed response below and would be grateful if the reviewer could consider increasing their rating if they found our response convincing.
> ***
> > I'm curious whether the proposed search algorithm is suitable algorithm to search in the reduced search space. I think ablation on search algorithm is necessary to make sure that the algorithm is one of the contribution or not.
>
> We would like to emphasize we did *not* claim the search method described as a key contribution. As we discussed in the Introduction, our focus is primarily on search space and its importance (“Line 090: While previous works have attempted to alleviate this problem by improving the search strategy, search space design has been largely overlooked.”), which is supported by our findings in Sec 3 that after search space pruning, even “randomly sampled discrete prompts can be competitive with the state of the art” (Line 283). As we stated in Line 355, we deliberately opted to use a very basic algorithm to contrast with previous methods with advanced search algorithms but made little to no attempt to improve the search space.
>
> It is indeed possible (and in some tasks indeed the case — see our response later) that, as the reviewer suggested, another search algorithm is superior to the one adopted in the paper. However, if that is the case, the fact that the suboptimal search algorithm used in this paper with intelligent search space pruning outperforms complicated, state-of-the-art search algorithm without search space pruning exactly proves the point of this paper, that search space is equally, if not more important, which is a key aspect the community has so far (largely) ignored. We will show later, per the reviewer’s suggestions, that it is indeed true that CLaPS benefits from search method improvements, leading to an even larger gain.
> ***
> > The experiments are only conducted on Flan-T5 models.
>
> We acknowledge the reviewer’s concerns. The chief reason we focused on Flan-T5 models is that of the model scale that we have computational resources to perform extensive experiments for, instruction-finetuned Flan-T5 achieves the best performance. We will endeavor to include results on more models in the future.
> ***
> > The number of queries to the target model should be considered as the evaluation metric for resource usage.
> > Test accuracy vs number of queries plot of ClaPS and other black-box prompt optimization method?
>
> We thank the reviewer for suggesting also considering the number of queries as a metric for resource usage. We now update our Table 3 by also including the number of queries. By comparing with the strongest baseline, RLPrompt, the proposed ClaPS method is more query efficient than RLPrompt by leveraging half the number of queries with better task performance.  Though we would like to provide the visualization of the plot of test accuracy vs. the number of queries, we are not allowed to attach any links. However, Figure 1 can still be interpreted with the number of queries as each method requires a fixed time span per query. Therefore, with the support of updated Table 3 and Figure 1, ClaPS also shows as the most query-efficient method, and other baselines (e.g. the blue line, RLPrompt) will not surpass ClaPS given even larger query budgets. We will include this number of query plot as the reviewer suggested in the camera-ready version.
>
> | Method             |#Params|VRAM| Time| #Query  | SST-2(%)  |
>  |---------------------|----------|----------|----------|----------|----------|
> | FT                    | 250M  |6.53GB|0.42min|     -       |76.15|
> | RLPrompt        |  3M     |3.60GB|65.1min| 12000  |86.01|
> | BDPL               | 1K      |2.54GB|0.20min|   600   |84.89|
> | Search             |   0       |2.54GB|1.26min|  4000  |85.82|
> | Prune&Search |   0       |2.54GB|7.65min| 24000 |87.84|
> | ClaPS              |   0       |2.54GB|1.80min|  6000  |87.78|
>
> ***
> > Discrete prompt optimization problem is a combinatorial optimization problem. In ClaPS, the authors propose search algorithm based on evolutionary method. However, the effectiveness of the proposed search method was not exploited enough.
>
> We thank the reviewer for the insightful suggestions; we tested some of the suggestions (but not all of them due to time and resource constraints during rebuttal), and we indeed found some of the suggested algorithms perform well – we will include full sets of such results for the camera-ready version.
>
> Re-emphasizing the point made above, though, we’d still like to stress the fact that the primary focus of this paper is on search space, not search methods and that the key contributions of the paper stand regardless of whether an alternative search algorithm outperforms, although we do thank the reviewer for suggesting alternative, potentially more suitable search algorithms that have made the performance of CLaPS even stronger.
>
> Specifically, in the table below, we added a greedy search algorithm and a particle swarm optimizer as suggested. The greedy algorithm starts with an empty string, iterates through the pruned search space (200 tokens in our case), and picks the token that leads to the highest reward (or lowest loss). Conditional on this token, we then pick the second token that, when combined with the first chosen token, led to the highest reward. This process continues until all 5 tokens are selected – this process requires $k|\mathcal{V}|$ total model queries, and thus shrinking $|\mathcal{V}|$ leads to an $\mathcal{O}(n)$ speedup.
>
> The particle swarm optimizer is adapted from the [open-source implementation](https://github.com/thunlp/SememePSO-Attack) of the paper [3] that the reviewer mentioned. The main changes made are to reflect the fact that the prompting setup is less restrictive than the adversarial attack, which would also require the changes in text to be as imperceptible as possible. As such, we removed constraints such as only substitution is allowed, and, unlike adversarial attacks where the algorithm is terminated whenever a successful perturbation is found, we always run full 40 epochs.
>
> We will also add the rest algorithms as the reviewer suggested in camera-ready. We hope the reviewer would sympathize with us that these algorithms are significantly more complicated given the short rebuttal timeframe.
>
> | Method    | RLP.  | Genetics (Ours) | Greedy (Ours) |Particle Swarm (Ours) |
> |--------------|---------|------------------------|--------------------|------------------------------|
> | SST-2      |86.01+-1.32   |87.78+-1.77 |**90.37**            |87.55+-1.81|
> | RTE        |78.52+-0.60   |**81.23+-0.56**  |79.42            |79.93+-1.85|
> | SNLI       |63.06+-0.42   |65.92+-3.14  |63.47            |**66.12+-3.35**|
> | QNLI       |74.85+-3.40   |70.52+-2.04 |**80.07**             |69.45+-1.73|
> | MNLI       |**57.60+-0.88**  |50.45+-3.09  |57.02             |53.71+-1.78|
> | MRPC     |58.82+-1.83  |68.43+-3.03  |65.93             |**70.83+-2.14**|
> | QQP       |80.09+-0.59   |80.87+-0.75 |81.40             |**81.51+-0.25**|
> | ag_news |**77.20+-0.76**   |76.06+-0.85 |77.13             |77.03+-1.13|
> | Average    |71.98           |72.66            |**74.35**             |73.27          |
>
>
> ***
> > Since ClaPS is a black-box prompt optimization method, you may apply the proposed method into black-box API such as GPT-4 or ChatGPT. Can you provide comparison between manual prompt and discrete prompt optimized by ClaPS?
>
> We use the popular definition of “black box” following published works like RLPrompt [1], BDPL [2], and earlier works like BBT [3] that while no model architecture, parameter, or gradient is required, the output probabilities over the verbalizer classes are still expected. However, to our knowledge, the public GPT-4 or ChatGPT APIs do not provide the logits. However, as we responded previously, we appreciate the reviewer’s feedback and will endeavor to include experiments on other models of similar scale to the GPT models to show the point further.
>
> [1] Deng, M., Wang, J., Hsieh, C. P., Wang, Y., Guo, H., Shu, T., ... & Hu, Z.. Rlprompt: Optimizing discrete text prompts with reinforcement learning. EMNLP 2022.
>
> [2] Diao, S., Huang, Z., Xu, R., Li, X., Lin, Y., Zhou, X., & Zhang, T. Black-box prompt learning for pre-trained language models. TMLR 2022.
>
> [3] Sun, T., Shao, Y., Qian, H., Huang, X., & Qiu, X. Black-box tuning for language-model-as-a-service. ICML 2022
> ***
> > Does the resource (time and vram) in table 3 contain the resource used in clustering and pruning step?
>
> The table does not include the clustering step, which is a preprocessing step with negligible computation cost on the CPU and is done *once for all tasks per model*.
>
> The table does contain the cost for both the pruning phase and search phase, and the time difference between Search and ClaPS is the time required for pruning. We will clarify this point when we revise the manuscript.

---

### Official Review · Reviewer_4AgN · 2023-08-05

**Soundness:** 4

**Excitement:**

3: Ambivalent: It has merits (e.g., it reports state-of-the-art results, the idea is nice), but there are key weaknesses (e.g., it describes incremental work), and it can significantly benefit from another round of revision. However, I won't object to accepting it if my co-reviewers champion it.

**Paper Topic And Main Contributions:**

This paper addresses the problem of a large search space in prompt search. To reduce the cost in searching over a large size of vocabulary, the authors propose an evolutionary algorithm combined with a vocabulary pruning method. The experimental results in the paper show that the proposed method outperforms state-of-the-art prompt search methods on 5-token prompt search with various NLP benchmarks.

**Questions For The Authors:**

A. I have some concerns about the design choices of the methods

(i) How did you set the values of the hyper-parameters including the number of clusters, pruning ratio, and number of evolution iterations?

(ii) How the performace correlates with each of hyper-parameters?

(iii) How sensitive is the performance to the hyper-parameters?

(iv) Why is the proposed method only evaluated in the settings of fixed 5-token prompts? I think that the proposed method have more scalability and flexibility to the prompt length than RL-based methods. So, would you provide some results with more diverse lengths (e.g., 2-token prompts, 10-token prompts, or dynamic-length prompts)?

(v) As the k-means clustering and evolutionary search have randomness, the reported results might not be replicated in other runs with different random seeds. Would you report the mean and variance of the benchmark performance?

B. Previous work [1] on prompt search has shown that the searched prompts might include some unreadable words, but those by the proposed method seems mostly readable and semantically related to given tasks. What component of the proposed method does make the differences?

[1] Deng et al., RLPrompt: Optimizing Discrete Text Prompts with Reinforcement Learning, EMNLP 2022.

---

I have read the response. The authors have mostly addressed my concerns. Thus, I adjust the scores in favor of acceptance.

**Reasons To Accept:**

A. The paper is easy-to-follow. The emprical observations in the paper clearly illustrate the problem.

B. The proposed method is simple, intuitive, and apparently sound.

C. The proposed method shows impressive results in terms of accuracy and efficiency.

**Reasons To Reject:**

A. The lack of analysis on the design choices raises the question about the optimality and robustness of the proposed method.

B. The proposed method is only evaluated on text classification tasks. It is unclear that the proposed method universally works well including reasoning and generation tasks. Although the authors partially mentioned this point in the limitation section in the paper, but it needs to be clarified whether the efficiency comes at a limited capability of models.

C. The paper lacks discussion about the differences of the prompts searched by the proposed method compared with those of existing methods.

**Reproducibility:**

4: Could mostly reproduce the results, but there may be some variation because of sample variance or minor variations in their interpretation of the protocol or method.

**Reviewer Confidence:**

3: Pretty sure, but there's a chance I missed something. Although I have a good feel for this area in general, I did not carefully check the paper's details, e.g., the math, experimental design, or novelty.

---

> ### Author Rebuttal · Authors · 2023-08-29
>
> We thank the reviewer for their detailed and constructive feedback! Please see below for our response. We hope that the reviewer will consider increasing their scores if they think we have sufficiently addressed their concerns.
> ***
> > A. The lack of analysis on the design choices raises the question about the optimality and robustness of the proposed method.
>
> Before responding to the reviewer’s questions on optimality and robustness of hyperparameters, we would like to stress that we **deliberately did not perform extensive hyperparameter tuning** for our method, nor did we seek to use a different, optimal set of hyperparameters for each task/model that many previous works did – while it is indeed possible to achieve so and to perform hyperparameter optimization per-task for even higher performance with our proposed method, we opted not to do so as this is typically expensive or otherwise unrealistic for real-life tasks. Instead, we achieved impressive results, as the reviewer acknowledged, with a single set of hyperparameters, without tuning, for all experiments. For the baselines, we did lean on the hyper-parameters suggested by previous research (which puts the baselines in a slight prior advantage compared to our method). We believe this is extremely strong evidence demonstrating the robustness and practicality of our method and is one of the key strengths of the proposed CLaPS method.
>
> With this key point in mind, we now respond to the rest of the questions in detail.
> ***
> > (i) How did you set the values of the hyper-parameters including the number of clusters, pruning ratio, and number of evolution iterations?
>
> As discussed, the hyperparameters are simply set to reasonable values without further tuning. As described in Fig 4 of our manuscript, the high-level rationale is to reduce the number of “surviving” vocabulary by an order of magnitude in each stage. Starting with $\mathcal{O}(10^4)$ tokens, we use clustering to reduce the vocabulary to $\mathcal{O}(10^3)$, pruning to $\mathcal{O}(10^2)$ and search to identify the final tokens to be used as discrete prompts.
>
> Specifically, the number of clusters is selected to be as large as computationally tractable to ensure diversity; evolution iteration is set at an empirically observed value until the accuracy stops increasing for several epochs. We already analyzed the effect of the pruning ratio in Fig 5, and we found the method to be quite robust – we can see that any non-zero pruning value leads to improvements over the performance at 0.0 (i.e., no pruning).
> ***
> >(ii) How the performace correlates with each of hyper-parameters? (iii) How sensitive is the performance to the hyper-parameters?
>
> **Number of evolution epochs**: The performance with respect to evolution epochs can be interpreted from Figure 1, where ClaPS starts from the first epoch and ends at the 30th epoch. The curve is observed to plateau around or before this point and is thus set as a common stop point for all our experiments.
>
> **Number of clusters**: Recalling that the purpose of clustering is to obtain a diverse set of candidate tokens and thus to save computation in the pruning stage, we now attach an ablation study that conducts the ClaPS pipeline with different numbers of clusters, and we will include this study in the appendix section. From the following table, having our number of clusters at 2000 stands as a good trade-off point for saving the cost while showing strong task performance.
>
> | #Cluster  | 20000 | 6000  | 2000   | 1000  |
> |--------------|----------|---------|----------|-----------|
> | SST-2      |87.84+-2.11|87.66+-2.07|87.78+-1.77|88.19+-0.52|
> | RTE         |81.95+-0.94|80.43+-1.57|81.23+-0.56|77.91+-1.43|
>
> **Pruning ratio**: The most sensitive hyperparameter is the pruning ratio that controls the size of the search space, and we have included relevant discussions in Figure 5 and Line 559-570.
> ***
> > (iv) Why is the proposed method only evaluated in the settings of fixed 5-token prompts? I think that the proposed method have more scalability and flexibility to the prompt length than RL-based methods. So, would you provide some results with more diverse lengths (e.g., 2-token prompts, 10-token prompts, or dynamic-length prompts)?
>
> The reviewer is correct that our method is more flexible, and we used fixed 5-token prompts as we would like a direct comparison with the strongest baseline of RLPrompt, which achieved the best performance in the 5-token setup. We now provide an ablation study with various token lengths, and we will include this result in the appendix section. First, increasing the token length from 2 to 5 leads to an improvement in performance over the two tasks. This is expected as a longer sentence can provide more expressive control and description for prompting the language model. By further increasing the token length from 5 to 10, we observe a decrease in performance and consider this to be due to the dimensionality problem in derivative-free optimization.
>
> | #Token     | 2               | 5                   | 10            |
> |--------------|----------------|------------------|--------------|
> | SST-2      |87.04+-1.18|87.78+-1.77  |87.41+-0.88|
> | RTE         | 80.07+-1.87|81.23+-0.56 |79.49+-1.59|
> ***
> > (v) As the k-means clustering and evolutionary search have randomness, the reported results might not be replicated in other runs with different random seeds. Would you report the mean and variance of the benchmark performance?
>
> In the main results, Tables 1 and 2, we have reported the mean for 5 random seeds for the Flan-T5 base model, and the standard deviation was omitted for conciseness. We recall that our Figure 1 and Figure 3 are also reported with 5 random seeds, which ensured the reliability of our observed performance improvements. We now attach Table 1 with standard deviations, and we will include this new table in the appendix section. The reviewer is also referred to our codebase in the submitted supplementary materials for any reproducibility concerns. We will also open-source our codebase.
>
> | Method | FT                | Manual  | BDPL       | RLP.            | Search      | ClaPS        |
> |--------------|----------------|----------|-----------------|----------------|----------------|----------------|
> | SST-2      |76.19+-0.93|85.32   |84.89+-2.29|86.01+-1.32|85.85+-1.79|87.78+-1.77|
> | RTE         |51.55+-2.46|73.65   |72.27+-2.37|78.52+-0.60|77.47+-1.13|81.23+-0.56|
> | SNLI        |60.98+-4.18|48.97   |51.65+-3.95|63.06+-0.42|59.30+-2.27|65.92+-3.14|
> | QNLI       |67.94+-4.19|62.40   |61.53+-1.81|74.85+-3.40|65.83+-2.20|70.52+-2.04|
> | MNLI       |45.99+-4.81|43.15   |42.52+-4.29|57.60+-0.88|47.30+-3.85|50.45+-3.09|
> | MRPC     |68.73+-0.71|69.12   |71.49+-7.93|58.82+-1.83|72.74+-1.11|68.43+-3.03|
> | QQP        |66.31+-2.81|79.07   |68.44+-2.64|80.09+-0.59|80.57+-1.76|80.87+-0.75|
>
> ***
> > B. The proposed method is only evaluated on text classification tasks. It is unclear that the proposed method universally works well including reasoning and generation tasks. Although the authors partially mentioned this point in the limitation section in the paper, but it needs to be clarified whether the efficiency comes at a limited capability of models.
>
> We mainly follow the evaluation datasets from competitive baselines like BDPL and RLPrompt. In addition, we also carry out more challenging experiments on multilingual XNLI datasets to reveal the potential of different prompt search methods in multilingual learning. We will endeavor to include results on more models and datasets as part of future work.
> ***
> > C. The paper lacks discussion about the differences of the prompts searched by the proposed method compared with those of existing methods.
>
> In Lines 543-558, we provide a short discussion of the ClaPS-discovered prompts. Since the ClaPS is not explicitly tuned towards fluency, we list ClaPS-discovered prompts as an interesting reference for readers. We now attach a table below that contains the prompts from the strongest baseline. We will include the corresponding discussion in the appendix section in the camera-ready version.
>
> It is noticeable that ClaPS shares some words with competitive baselines, and these words (e.g., 'review' and 'answer') are usually 'influential prompts' identified by our pruning strategy and have significant impacts on the model’s prediction.  With a similar or even better quality of prompts, ClaPS stands out by first establishing an efficient search space while saving significant computation costs. For a more detailed discussion on the readability of our prompts, please find our response to the next question.
>
> |Method     |SST-2  | RTE    |
> |--------------|----------|----------|
> | RLP.      |ReviewCustomerBankBankBank  | DatabaseansweranswerYesĠyes|
> | ClaPS   |cruise perfect properly review cruise |answer respectively minimum tell answer|
>
> ***
> > B. Previous work [1] on prompt search has shown that the searched prompts might include some unreadable words, but those by the proposed method seems mostly readable and semantically related to given tasks. What component of the proposed method does make the differences?
>
> As described in Appendix A.1 Lines 821-825, we ensure the pruning phase works on a single word while filtering all special tokens, which eventually improves the readability of the whole sentence with multiple words. In addition, during the pruning phase, we discover that the most influential prompts are usually contextual words or in-domain words, e.g., 'review' and 'answer' for SST-2 and RTE, respectively. This contextual feature also helps the interpretability of our results. RLPrompt has no implementation of search space reduction, which occasionally results in unreadable and gibberish prompts. This finding further coincides with the main claim of this paper, and it emphasizes the importance of the search space for prompts.

---

### Official Review · Reviewer_a6rh · 2023-08-05

**Soundness:** 3

**Excitement:**

3: Ambivalent: It has merits (e.g., it reports state-of-the-art results, the idea is nice), but there are key weaknesses (e.g., it describes incremental work), and it can significantly benefit from another round of revision. However, I won't object to accepting it if my co-reviewers champion it.

**Paper Topic And Main Contributions:**

The paper introduces ClaPS, a Clustering and Pruning method for Efficient Black-box Prompt Search, to enhance prompt-based learning for large pretrained language models (LLMs). By identifying and focusing on influential prompt tokens, ClaPS achieves state-of-the-art performance across tasks and LLMs while reducing search costs, highlighting the significance of search space design and optimization for efficient black-box prompt-based learning.




**Questions For The Authors:**

See above.

**Reasons To Accept:**


- The paper presents an interesting and important research problem, addressing the efficiency of black-box prompt search for large pretrained language models (LLMs).

- It showcases strong performance compared to recent baselines, demonstrating the effectiveness of the proposed Clustering and Pruning for Efficient Black-box Prompt Search (ClaPS) method across various tasks and LLMs.

- Figure 4 provides a clear and easy-to-understand visualization of the sensitivity analysis and influential prompt tokens, contributing to the paper's overall clarity and readability.

**Reasons To Reject:**

1. The proposed Clustering and Pruning for Efficient Black-box Prompt Search (ClaPS) method involves several stages, which may lead to reduced elegance and lack of principled reasoning. This complexity could make it challenging to understand and reproduce the method in a straightforward manner.

2. The paper lacks comprehensive implementing details of the ClaPS method.

3. The paper does not report standard deviations or conduct statistical tests when comparing the proposed method with the baselines which makes it difficult to assess the significance and reliability of the observed performance improvements.

**Reproducibility:**

3: Could reproduce the results with some difficulty. The settings of parameters are underspecified or subjectively determined; the training/evaluation data are not widely available.

**Reviewer Confidence:**

3: Pretty sure, but there's a chance I missed something. Although I have a good feel for this area in general, I did not carefully check the paper's details, e.g., the math, experimental design, or novelty.

---

> ### Author Rebuttal · Authors · 2023-08-29
>
> We thank the reviewer for their detailed and insightful review! Please see our response below, which we believe has addressed all the concerns. We hope that in light of the response, the reviewer could consider improving their score.
> ***
> > The proposed Clustering and Pruning for Efficient Black-box Prompt Search (ClaPS) method involves several stages, which may lead to reduced elegance and lack of principled reasoning. This complexity could make it challenging to understand and reproduce the method in a straightforward manner.
>
> We emphasize that while ClaPS involves several stages, the design of each and every stage is strongly justified with technical rationales and motivated in the paper rather than artificially constructed for unnecessary complexity. We thus do not see as a weakness the fact that our method involves multiple stages. In a nutshell, 1) we perform clustering to reduce the computational cost (note that the method does work even without the clustering step, 2) pruning to curate a high-performing search space, and 3) search to identify a final set of token prompts. The reviewer is also welcome to run our attached code, which, in our opinion, alleviates any potential challenges the reviewer mentioned regarding reproducibility.
> ***
> > The paper lacks comprehensive implementing details of the ClaPS method.
>
> We report the overall step-by-step walkthrough in Alg. 1, and all our implementation details are in Section 3 and Appendix A.1. We also kindly refer the reviewer to our codebase in the submitted supplementary materials to examine all the implementation and reproducibility details. We will also open-source our codebase, as stated in the paper.
> ***
> > The paper does not report standard deviations or conduct statistical tests when comparing the proposed method with the baselines, which makes it difficult to assess the significance and reliability of the observed performance improvements.
>
> We thank the reviewer for pointing this out. In the main results, Tables 1 and 2, we have reported the mean for 5 random seeds for the Flan-T5 base model, and the standard deviation was omitted for conciseness. We recall that our Figure 1 and Figure 3 are also reported with 5 random seeds, which ensured the reliability of our observed performance improvements. We now attach Table 1 with standard deviations, and we will include this new table in the appendix section to remove any concerns.
>
>
> | Method | FT                | Manual  | BDPL       | RLP.            | Search      | ClaPS        |
> |--------------|----------------|----------|-----------------|----------------|----------------|----------------|
> | SST-2      |76.19+-0.93|85.32   |84.89+-2.29|86.01+-1.32|85.82+-1.79|87.78+-1.77|
> | RTE         |51.55+-2.46|73.65   |72.27+-2.37|78.52+-0.60|77.47+-1.13|81.23+-0.56|
> | SNLI        |60.98+-4.18|48.97   |51.65+-3.95|63.06+-0.42|59.30+-2.27|65.92+-3.14|
> | QNLI       |67.94+-4.19|62.40   |61.53+-1.81|74.85+-3.40|65.83+-2.20|70.52+-2.04|
> | MNLI       |45.99+-4.81|43.15   |42.52+-4.29|57.60+-0.88|47.30+-3.85|50.45+-3.09|
> | MRPC     |68.73+-0.71|69.12   |71.49+-7.93|58.82+-1.83|72.74+-1.11|68.43+-3.03|
> | QQP        |66.31+-2.81|79.07   |68.44+-2.64|80.09+-0.59|80.57+-1.76|80.87+-0.75|

---

### Meta-Review · Area_Chair_NJ96 · 2023-09-18

**Recommendation:** 4

**Metareview:**

The paper presents a clustering and pruning method for black-box prompt search called CLaPS. The methodology integrates evolutionary algorithms with vocabulary pruning techniques to address the extensive search space associated with prompt search. The approach appears to outperform existing black box optimization methods in wall clock time while achieving similar or better prompts.

Some reviewers felt that the multi-stage nature of ClaPS potentially makes the approach a bit convoluted, hindering straightforward understanding and reproduction. Figure 4 does a reasonable job of visually demonstrating the different stages in the algorithm. Similarly, some reviewers found that the method lacked enough details/justifications for each step. During the rebuttal, the authors argue that their main contribution is the use of pruning to reduce the search space, and that a better search algorithm may further improve performance.

More qualitative comparison on how the prompts derived using ClaPS differ from those of existing methods. During the rebuttal, the authors provide additional comparison and commentary on the differences between the CLaPS output and RLPrompt output. However, this seems like a clear place where the work could be further improved.

---

### Decision · Program_Chairs · 2023-10-07

**Decision:**

Accept-Findings

**Comment:**

The paper presents a clustering and pruning method for black-box prompt search called CLaPS. The methodology integrates evolutionary algorithms with vocabulary pruning techniques to address the extensive search space associated with prompt search. The approach appears to outperform existing black box optimization methods in wall clock time while achieving similar or better prompts.

Some reviewers felt that the multi-stage nature of ClaPS potentially makes the approach a bit convoluted, hindering straightforward understanding and reproduction. Figure 4 does a reasonable job of visually demonstrating the different stages in the algorithm. Similarly, some reviewers found that the method lacked enough details/justifications for each step. During the rebuttal, the authors argue that their main contribution is the use of pruning to reduce the search space, and that a better search algorithm may further improve performance.

More qualitative comparison on how the prompts derived using ClaPS differ from those of existing methods. During the rebuttal, the authors provide additional comparison and commentary on the differences between the CLaPS output and RLPrompt output. However, this seems like a clear place where the work could be further improved.